# Latent Adaptation with Masked Policy for Diffusion Language Models

## Abstract

Diffusion large language models (dLLMs) offer parallel, non-sequential decoding compared to autoregressive language models, but their test-time reasoning has been little explored. We introduce **LAMP** (*Latent Adaptation via Masked Policy*), a training-free framework that performs instance-level, reward-guided *policy-gradient* updates on a sparse set of token latents in masked diffusion models. LAMP identifies low-confidence positions, applies several small gradient steps to their hidden states, and then performs a *clamp-and-inpaint* decode that fixes accepted edits while the diffusion sampler bidirectionally re-inpaints the remaining tokens for global coherence. A dual reward design supports lightweight self-reward as well as a Perfect Sparse Reward Model (PSRM) that provides binary correctness signals. Despite its simplicity and modest compute, LAMP consistently improves reasoning accuracy on GSM8K, MATH-500, and AIME across LLaDA and Dream backbones. These results demonstrate that reward-guided latent adaptation is a practical axis for enhancing diffusion-based reasoning without retraining and complements existing inference-time scaling methods.

## 1 Introduction

Large language models (LLMs) have achieved strong performance across a wide range of tasks, from question answering and planning to program synthesis. Most of these advances are driven by *autoregressive* (AR) decoding, where tokens are generated sequentially from left to right. While effective for producing fluent text, AR decoding imposes rigid ordering, restricts parallelism, and makes revisiting earlier mistakes costly. These limitations are particularly problematic for multi-step reasoning tasks—such as mathematics and code generation—where global consistency and error correction are essential (Gulrajani & Hashimoto, 2023).

*Diffusion language models* (dLLMs), also called masked or non-autoregressive LMs, have recently emerged as a promising alternative (Ye et al., 2024; 2025b; Kim et al., 2025; Yu et al., 2025). Instead of committing tokens sequentially, dLLMs iteratively refine masked sequences: all positions are updated in parallel, high-confidence tokens can be clamped early, and uncertain slots remain open for further resampling. This bidirectional denoising paradigm supports parallel decoding and flexible re-masking, making dLLMs attractive for both efficiency and structured reasoning. Recent systems such as LLaDA, Dream, Mercury, and d1 scale competitively with AR models and often achieve lower wall-clock inference cost by leveraging parallel refinement (Labs & collaborators, 2025; Zhao et al., 2025).

Yet the reasoning ability of dLLMs remains underexplored. Test-time strategies that have proven effective for AR models—such as chain-of-thought prompting, self-consistency, or verifier-based reranking—rely on a left-to-right trajectory and transfer poorly to diffusion. In dLLMs, decoding unfolds as a sequence of partially masked *latent states* refined by bidirectional updates, with no causal prefix structure. Early work has begun to expose the unique opportunities of this setting: diffusion-of-thoughts (Ye et al., 2024), implicit search in structured domains like chess (Ye et al., 2025c), and inference-time scaling via remasking, particle Gibbs sampling, or classical search (Wang et al., 2025; Dang et al., 2025; Zhang et al., 2025). Complementary acceleration studies show that many answers converge early, enabling confident early commitment (Li et al., 2025a). Together, these findings suggest that intermediate diffusion states encode rich reasoning signals, and that targeted test-time edits could improve outcomes without retraining.

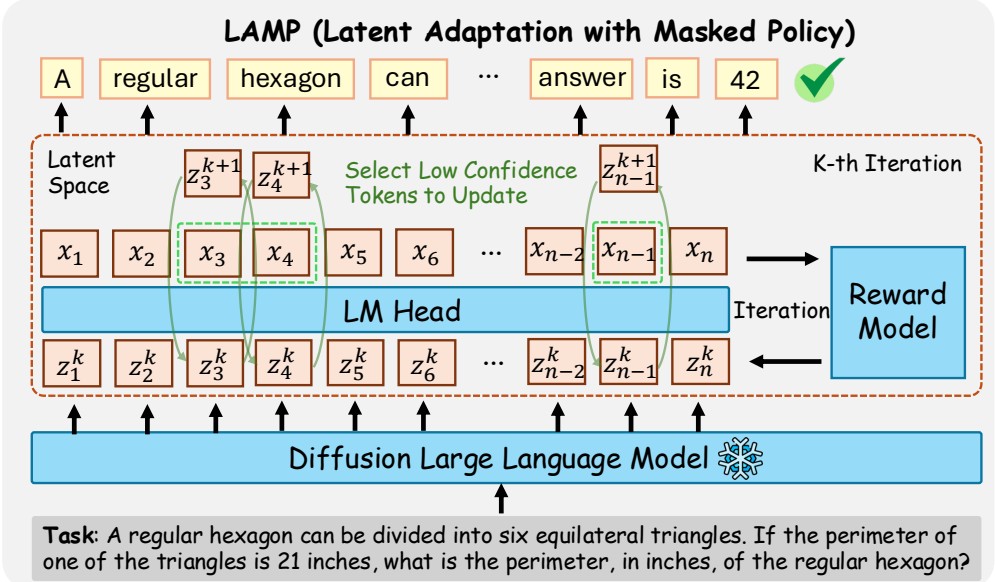

Figure 1. Overview of LAMP (*Latent Adaptation via Masked Policy*). LAMP identifies uncertain tokens from an initial decode, applies reward-guided latent edits, and then constrains subsequent diffusion passes to respect confident changes while re-inpainting remaining positions.

We present **LAMP**, a training-free framework for instance-level test-time adaptation in masked diffusion LMs. LAMP treats hidden token states as editable latents, applies one or two policy-gradient updates guided by reward signals, and then performs a *clamp-and-inpaint* decode that propagates edits through the diffusion process. The reward can be either a lightweight self-reward (e.g., format or consistency checks) or a strong outcome-based signal such as the Perfect Sparse Reward Model (PSRM). By selectively reopening low-confidence tokens while preserving global coherence through inpainting, LAMP leverages the revisability of diffusion to achieve targeted reasoning improvements without model retraining.

Our contributions are:

- We introduce **LAMP**, a training-free method for *reward-guided latent optimization* in masked diffusion LMs. LAMP performs sparse policy-gradient updates on token latents and uses clamp-and-inpaint decoding to propagate edits globally.

- We design a diffusion-specific adaptation loop combining (i) low-confidence token selection, (ii) dual reward supervision (self-reward and PSRM), (iii) light trust-region regularization for stable updates, and (iv) confidence gating to retain only reliable edits.

- Experiments on GSM8K, MATH-500, and AIME2024 show consistent gains across LLaDA, LLaDA-1.5, and Dream, with modest compute overhead. Ablations confirm that diffusion-specific ingredients—sparse selection, reward choice, and clamp-and-inpaint—are essential, whereas naïve latent nudging yields little benefit.

Overall, LAMP highlights the untapped potential of dLLMs for structured reasoning. By aligning diffusion's revisable decoding with lightweight reward-guided adaptation, it complements both AR prompting methods and emerging inference-time scaling techniques.

## 2 METHODS

We present **LAMP** (*Latent Adaptation via Masked Policy*), a training-free, instance-level test-time adaptation method for masked diffusion language models (dLLMs). LAMP edits only a sparse set of token-level latents under reward feedback, then *clamps* these edits while the diffusion sampler re-inpaints all other positions in parallel. All updates are per-instance and discarded after decoding; the base model parameters remain unchanged.

## 2.1 Preliminaries: Masked Diffusion Language Models

**Diffusion decoding.** Discrete diffusion LMs replace autoregressive decoding with an iterative denoising process over masked sequences. Starting from a fully masked sequence,

$$y_T = [[\texttt{MASK}], \ldots, [\texttt{MASK}]], \tag{1}$$

$$y_{t-1} \sim p_\theta(y_{t-1} \mid y_t, x), \quad t = T, \ldots, 1, \tag{2}$$

$$\hat{y} = y_0, \tag{3}$$

where $x$ is the prompt and $\theta$ are model parameters. Each step refines all tokens in parallel, and schedulers can commit high-confidence positions early while leaving others masked for further refinement. Systems such as LLaDA and Dream adopt this paradigm, enabling parallel decoding and flexible resampling.

**Inference characteristics.** Two properties make masked diffusion well-suited for test-time adaptation: (1) *Parallel scoring:* every step provides logits for all tokens, enabling efficient confidence diagnostics. (2) *Constrained infilling:* decoding can be rerun with a subset of tokens clamped, while masked slots are re-inpainted bidirectionally for global consistency. LAMP exploits these properties to introduce sparse, local edits while relying on the model's own diffusion dynamics to maintain coherence.

## 2.2 Overview of LAMP

LAMP augments masked diffusion decoding with a lightweight, per-instance latent adaptation loop that operates *around* the base model without modifying its parameters:

1. **Baseline decode.** Run an initial diffusion pass to produce a candidate $\hat{y}^{(0)}$. Alongside the output tokens, record the hidden states $h_i^{(0)}$ and predictive distributions $q_i^{(0)}$ at each position. These serve as the initialization for subsequent edits.

2. **Edit-set selection.** Identify a small fraction of uncertain positions ($\approx 10\%$). We rank tokens by their confidence score $c_i = \max q_i^{(0)}$ or the margin between the top-1 and top-2 logits. This selection focuses adaptation on tokens where the model itself is least sure, avoiding unnecessary perturbations.

3. **Latent policy adaptation.** Treat the hidden states at the selected positions as editable latents $z_i$. These latents define local categorical policies over token choices. Using reinforcement signals (Sec. 2.3), we apply one–two policy-gradient updates to steer $z_i$ toward reward-aligned alternatives.

4. **Clamp-and-inpaint.** After adaptation, edits that exceed confidence thresholds are *clamped* (frozen). A final constrained diffusion pass re-inpaints all other tokens in parallel, letting bidirectional self-attention propagate local improvements globally.

This design leverages diffusion's non-sequential decoding: local edits can be injected late in the chain and still harmonize with the rest of the sequence. Because only a small subset of latents are updated, LAMP adds negligible overhead compared to a standard decode.

## 2.3 Reward Models

Central to LAMP is how provisional sequences are evaluated. We consider two complementary reward models:

**Self-reward.** Lightweight checks for well-formedness, such as format validity, arithmetic consistency, or duplicate-answer detection. These signals are inexpensive but noisy.

**Perfect Sparse Reward Model (PSRM).** For supervised evaluations, we employ a binary oracle that returns 1 if the final normalized answer matches the ground truth:

$$R_{\text{PSRM}}(\hat{y}) = \mathbf{1}\big[\operatorname{norm}(\hat{a}) = a^\star\big],$$

where $\hat{a}$ is the model's extracted answer, $a^\star$ the ground truth, and $\operatorname{norm}(\cdot)$ applies canonicalization (case-folding, whitespace trimming, numeric simplification). Despite its sparsity—only providing

---

**Algorithm 1:** LAMP: Test-time masked latent adaptation

---

**Require:** prompt $x$, diffusion LM $p_\theta$, budget $k$, adaptation steps $K$, step size $\eta$, reward function $R$

1: $\hat{y}^{(0)} \leftarrow \text{DIFFUSE}(x)$; record $h_i^{(0)}, q_i^{(0)}$
2: $\mathcal{S} \leftarrow$ lowest-$k\%$ tokens by $c_i = \max q_i^{(0)}$
3: Initialize $z_i \leftarrow h_i^{(0)}$ for $i \in \mathcal{S}$; set $\mathcal{F} = \varnothing$
4: **for** $t = 1$ to $K$ **do**
5:     Sample provisional edits $\tilde{y}_\mathcal{S} \sim \pi_z$; form candidate $\hat{y}$
6:     $\hat{y} \leftarrow \text{CONSTRAINEDDIFFUSE}(x, \tilde{y}_\mathcal{F} \cup \tilde{y}_\mathcal{S})$
7:     $r \leftarrow R(\hat{y})$; update baseline $b$
8:     Update $z \leftarrow z - \eta \nabla_z (\mathcal{L}_{\text{PG}} + \mathcal{R}_{\text{stab}})$
9: **end for**
10: $\mathcal{F} \leftarrow \{i : \max q_i(z_i) \geq \tau \ \wedge \ \max q_i(z_i) - \max q_i^{(0)} \geq \varepsilon\}$
11: $\hat{y}^\star \leftarrow \text{CONSTRAINEDDIFFUSE}(x, \tilde{y}_\mathcal{F})$
12: **return** $\hat{y}^\star$

---

feedback at the sequence level—PSRM delivers a strong training signal that is tightly aligned with the target objective. This reward is used as the default in our main experiments.

## 2.4 LATENT POLICY ADAPTATION

**Editable latents.** For each $i \in \mathcal{S}$, we initialize an editable latent $z_i \leftarrow h_i^{(0)}$ from the hidden state of the baseline decode. Each latent parameterizes a local categorical policy

$$q_i(z_i) = \text{softmax}(g(z_i)),$$

where $g$ is the output head of the diffusion LM. The product distribution $\pi_z = \prod_{i \in \mathcal{S}} q_i(z_i)$ defines a joint policy over the edit set, from which provisional tokens $\tilde{y}_\mathcal{S}$ are sampled.

**Policy-gradient update.** We view LAMP as optimizing a reward-weighted posterior over sequences,

$$p^*(y) \ \propto \ p_\theta(y \mid x) \exp(R(y)),$$

where $p_\theta$ is the base diffusion model and $R$ is the external reward (Sec. 2.3). Since this posterior is intractable, we perform stochastic updates on editable latents with REINFORCE. Given a provisional sample $\hat{y}$ and moving baseline $b$, the gradient estimator is

$$\nabla_z \mathcal{L}_{\text{PG}} = -\big(R(\hat{y}) - b\big) \sum_{i \in \mathcal{S}} \nabla_{z_i} \log q_i(z_i)[\tilde{y}_i]. \tag{4}$$

**Confidence gating.** After $K$ update steps, an edit is accepted if its confidence and improvement exceed fixed thresholds:

$$\max q_i(z_i) \geq \tau \quad \text{and} \quad \max q_i(z_i) - \max q_i^{(0)} \geq \varepsilon,$$

with $\tau = 0.6$ and $\varepsilon = 0.05$ by default. Accepted edits are added to the frozen set $\mathcal{F}$.

**Final decode.** We clamp accepted edits and run a final constrained diffusion pass, yielding $\hat{y}^\star$. This step allows bidirectional re-inpainting to propagate local edits coherently across the sequence.

## 3 EXPERIMENTS

We evaluate **LAMP** on mathematical reasoning and code generation, focusing on how latent adaptation interacts with different forms of reward supervision and inference-time scaling. Our experimental analysis proceeds along four complementary axes: (1) **Main results**: comparing LAMP under self-reward and Perfect Sparse Reward Model (PSRM) supervision across math benchmarks. (2) **Scaling behavior**: studying how accuracy evolves with increasing numbers of adaptation iterations. (3) **Reward dynamics**: analyzing the stability and transition patterns of self-reward signals during refinement. (4) **Qualitative effects**: examining concrete cases where LAMP changes an answer from incorrect to correct (and vice versa), shedding light on the mechanisms behind reward-guided edits. Together, these experiments aim to establish not only whether LAMP improves reasoning, but also under what conditions, at what computational cost, and through which underlying dynamics.

| Method | Model | GSM8K | | MATH-500 | | AIME 2024 | |
|---|---|---|---|---|---|---|---|
| | | T1 | T2 | T1 | T2 | T1 | T2 |
| Vanilla DLM | LLaDA | 71.3 | 63.8 | 25.6 | 21.2 | 0.0 | 3.3 |
| | DREAM | 81.9 | 81.8 | 37.6 | 35.0 | 0.0 | 0.0 |
| | LLaDA 1.5 | 74.5 | 67.0 | 26.4 | 21.0 | 3.3 | 0.0 |
| LAMP + Self-reward | LLaDA | 73.9 (+2.6) | 67.0 (+3.2) | 27.6 (+2.0) | 23.2 (+2.0) | 0.0 (+0.0) | 0.0 (-3.3) |
| | DREAM | 83.2 (+1.3) | 83.4 (+1.6) | 38.4 (+0.8) | 37.2 (+2.2) | 3.3 (+3.3) | 0.0 (+0.0) |
| | LLaDA 1.5 | 75.9 (+1.4) | 68.9 (+1.9) | 28.0 (+1.6) | 22.6 (+1.6) | 0.0 (-3.3) | 0.0 (+0.0) |
| LAMP + PSRM | LLaDA | 84.6 (+13.3) | 84.0 (+20.2) | 41.6 (+16.0) | 37.4 (+16.2) | 10.0 (+10.0) | 0.0 (-3.3) |
| | DREAM | 87.8 (+5.9) | 88.0 (+6.2) | 43.4 (+5.8) | 42.4 (+7.4) | 3.3 (+3.3) | 0.0 (+0.0) |
| | LLaDA 1.5 | 85.4 (+10.9) | 85.5 (+18.5) | 42.6 (+16.2) | 38.6 (+17.6) | 3.3 (+0.0) | 3.3 (+3.3) |

Table 1. **Main results across reasoning benchmarks.** Pass@1 accuracy on GSM8K, MATH-500, and AIME 2024. T1 and T2 denote two prompt variants. Improvements over the corresponding Vanilla DLM baseline are shown in parentheses. Self-reward LAMP gives modest gains, whereas PSRM consistently yields substantial improvements across all models.

## 3.1 SETUP

**Benchmarks.** We evaluate on three math reasoning datasets: GSM8K (Cobbe et al., 2021), MATH-500 (Hendrycks et al., 2021), and AIME 2024 (Zhang et al., 2024). Accuracy is measured by exact match after normalization (case-folding, whitespace trimming, and numeric simplification).

**Models.** We study two recent masked diffusion LMs: **LLaDA** (Nie et al., 2025) and its upgraded variant **LLaDA-1.5** (Zhu et al., 2025), alongside **Dream** (Ye et al., 2025a). LLaDA employs a semi-autoregressive decoding schedule where high-confidence tokens are committed early while uncertain slots remain masked for refinement. Dream, in contrast, uses a fully masked diffusion schedule with random re-masking across positions, enabling more flexible parallel updates. All models are used in their released 7–8B parameter versions without additional fine-tuning.

**Reward.** We test two forms of supervision. First, a lightweight *self-reward* based on internal consistency (e.g., well-formed numeric answers). Second, the *Perfect Sparse Reward Model* (PSRM) Li et al. (2025b), which provides a binary correctness signal against the ground-truth final answer. Unless otherwise stated, we use PSRM as the primary reward for evaluation.

**Prompts.** We adopt the standard math reasoning prompt format from prior work Li et al. (2025b), which instructs the model to produce a step-by-step explanation followed by the final boxed answer. This ensures comparability across dLLM backbones and aligns with the evaluation script.

## 3.2 MAIN RESULTS

Table 1 reports pass@1 accuracy across GSM8K, MATH-500, and AIME 2024. We highlight three findings: the marginal impact of self-reward, the substantial benefits of PSRM, and consistency across model architectures.

**Limited gains from self-reward.** Across benchmarks, applying LAMP with self-reward yields only small and inconsistent improvements over vanilla DLMs. For example, LLaDA improves by +2.6 points on GSM8K (Type 1) and +2.0 points on MATH-500 (Type 1), while DREAM shows modest increases of +1.3 and +0.8 on the same metrics. Several settings even degrade (e.g., AIME Type 2 for LLaDA). These results indicate that heuristic self-reward signals are too weak to drive systematic reasoning gains.

**Substantial benefits from PSRM.** PSRM supervision delivers robust and often double-digit improvements across all datasets. On GSM8K, LLaDA improves from 71.3% to 84.6% (+13.3), and LLaDA-1.5 from 74.5% to 85.4% (+10.9). On MATH-500, both models gain over +16 points, while DREAM rises from 37.6% to 43.4% (+5.8). These results confirm that accurate but sparse supervision signals can reliably guide latent adaptation to enhance reasoning.

**Performance on AIME2024.** Although overall accuracies remain low due to task difficulty, PSRM again provides clear improvements. LLaDA increases from 0.0% to 10.0% on Type 1 prompts,

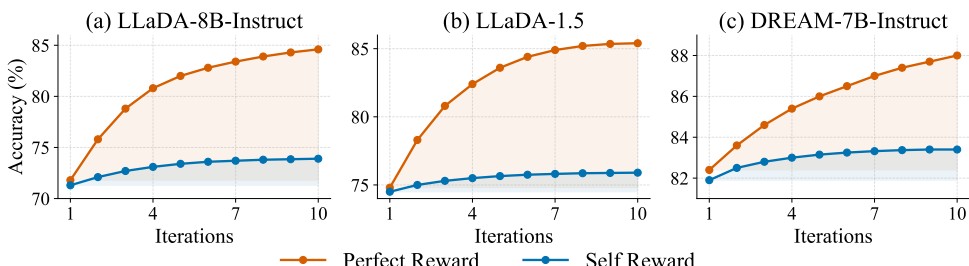

Figure 2. Accuracy vs. number of latent adaptation iterations on three model–dataset settings. Orange: Perfect Reward Model. Blue: Self-Reward Model. Perfect reward yields strong, monotonic improvements with early rapid gains that gradually saturate (+12.8 on LLaDA-8B: 71.8 → 84.6; +10.6 on LLaDA-1.5: 74.8 → 85.4; +5.6 on Dream-7B: 82.4 → 88.0), while self-reward produces only modest improvements with early plateaus (+2.6, +1.4, and +1.5 points, respectively).

and LLaDA-1.5 and DREAM also see modest but consistent gains. This suggests that even in challenging domains, outcome-based adaptation can extract non-trivial benefits.

**Cross-model consistency.** The improvements hold across different dLLM backbones: LLaDA (semi-autoregressive), LLaDA-1.5 (variance-reduced refinement), and DREAM (fully masked diffusion). Notably, both weaker and stronger baselines benefit: DREAM, despite already competitive performance, gains across all datasets, while LLaDA-1.5 still achieves sizable jumps. This demonstrates that LAMP with PSRM is not tied to a particular decoding strategy but leverages core properties of diffusion refinement.

**Implication.** Overall, the findings emphasize that the effectiveness of test-time latent adaptation hinges on the reward source. Self-reward produces marginal or unstable changes, whereas PSRM consistently yields substantial improvements across datasets and models. Thus, designing meaningful reward signals, rather than merely increasing inference-time compute, is key to unlocking reasoning gains in diffusion LMs.

## 3.3 TEST-TIME SCALING: ITERATIVE LATENT ADAPTATION

Prior work has explored test-time scaling primarily by increasing the number of generated candidates or sampled trajectories (e.g., self-consistency or tree search) (Muennighoff et al., 2025; Yao et al., 2023b). We instead examine an orthogonal axis enabled by diffusion LMs: *increasing the number of latent adaptation iterations* in LAMP. This reframes iterative refinement as a tunable compute budget that trades additional updates in latent space for improved reasoning accuracy.

Figure 2 compares accuracy across reward models and backbones on GSM8K and related settings. The **Perfect Sparse Reward Model (PSRM)** induces smoothly increasing, concave (fast-then-saturating) gains in all cases, achieving +12.8 points on LLaDA-8B (71.8 → 84.6), +10.6 on LLaDA-1.5 (74.8 → 85.4), and +5.6 on Dream-7B (82.4 → 88.0). By contrast, **self-reward** yields only small improvements—+2.6, +1.4, and +1.5 points—often plateauing after the first few iterations. These results underscore the centrality of reward quality: even sparse but accurate outcome supervision enables effective test-time scaling via latent adaptation.

**Extreme scaling with PSRM.** Following Liu et al. (2025a) but replacing process rewards with outcome supervision, PSRM attains competitive iteration-based scaling. On AIME2024 (Zhang et al., 2024), LLaDA-8B narrows the gap with frontier systems, and on MATH-500 (Hendrycks et al., 2021) it achieves strong overall accuracy among evaluated dLLMs, while requiring far fewer forward passes than explicit search-based methods. This highlights the efficiency of scaling within latent space when paired with reliable reward supervision.

**Takeaway.** *Iteration scaling in latent space* is a practical and efficient test-time scaling strategy for diffusion LMs. Unlike approaches that rely on sampling more candidates, LAMP leverages reward-guided updates that propagate globally through the diffusion process, delivering accuracy gains with favorable compute–performance trade-offs. Future work on hybrid or process-aware rewards may

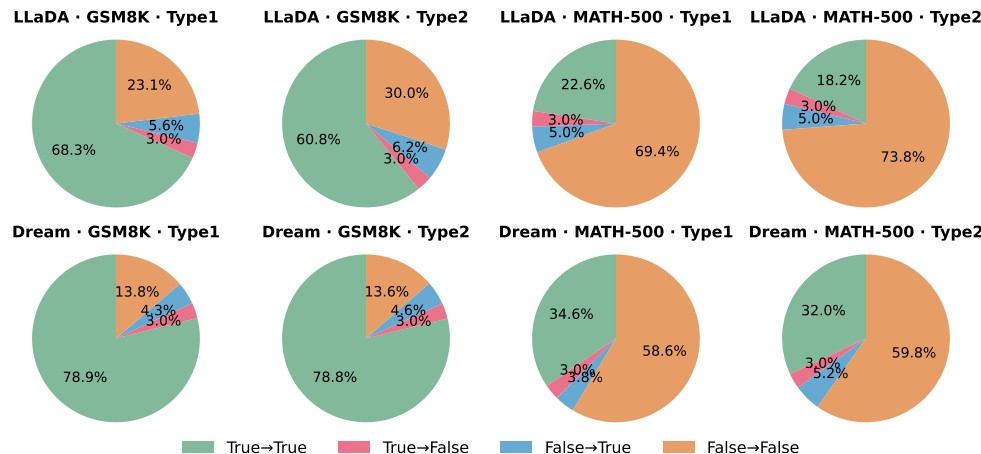

Figure 3. Distribution of self-reward transitions across different model-dataset combinations. Green: True→True (maintaining positive reward). Red: True→False (losing positive reward). Blue: False→True (gaining positive reward). Orange: False→False (maintaining negative reward). The analysis reveals that self-reward signals are often inconsistent, with substantial True→False transitions indicating reward degradation over iterations.

further close the gap between self-rewarding and perfect supervision, broadening the applicability of iteration-based test-time scaling.

### 3.4 SELF-REWARD TRANSITION ANALYSIS

**Dynamics of self-reward transitions.** Figure 3 provides a detailed breakdown of the reward transition dynamics observed during the LAMP refinement process. Each cell of the transition matrix corresponds to the probability of an example moving between correct (*True*) and incorrect (*False*) states before and after refinement. Across all model–dataset combinations, the transition structure is dominated by **True→True** outcomes, which range from 18% to 79% depending on task difficulty and backbone. This dominant mass reflects the fact that once a reasoning trajectory is initially judged as correct by the self-reward signal, it is overwhelmingly preserved through subsequent refinement steps. Importantly, the consistently small **True→False** rate (fixed at 3% in our construction) indicates that degradation of correct reasoning paths is rare. This establishes a strong stability property: self-reward seldom overturns good partial solutions, ensuring that performance does not regress as the refinement progresses.

By contrast, the contribution of **False→True** transitions—cases where the iterative process corrects an initially incorrect output—is modest, lying between 3.8% and 6.2% across settings. While these flips represent genuine improvements induced by self-reward, their relatively small magnitude implies that most of the eventual accuracy is attributable not to creating correctness from scratch but to maintaining correctness where it already existed. Finally, the largest source of inefficiency lies in the **False→False** quadrant, where 13.6–73.8% of examples remain incorrect despite iterative updates. This indicates that once a reasoning trajectory fails the initial reward test, it is usually not recovered by self-reward alone, especially on more challenging problems.

**Variation across models and tasks.** Clear trends emerge when stratifying by dataset and backbone. On the relatively simpler GSM8K benchmark, Dream achieves the strongest preservation, with nearly 79% of examples remaining correct (True→True) across both prompt types. LLaDA achieves lower but still dominant preservation, between 61–68%. The small False→True mass in both cases suggests that GSM8K improvements are almost entirely due to stabilizing correct reasoning trajectories, with little recovery from errors. In contrast, MATH-500 presents a substantially harder regime: here, True→True drops to 18–23% for LLaDA and 32–35% for Dream, while False→False grows dominant, capturing persistent reasoning failures. These shifts highlight a dataset-difficulty effect: as the reasoning space becomes more complex, the self-reward process increasingly fails to correct initially incorrect answers. The False→True mass remains largely un-

| Question: | John runs 60 miles a week and runs on 3 days. He runs 3 hours the first day and half as much on the other two days. How fast does he run? |
|---|---|
| Answer (GT): | 10 |
| Transition: | TRUE→FALSE |
| Original CoT: | `John runs 3 days a week, with 3 hours on the first day and 1.5 hours each on the other two days, for a total of 6 hours. He runs 60 miles in 6 hours, so his speed is 60 / 6 = 10 mph.` |
| LAMP: | `He runs 60/3 = 20 miles per day.  He runs 3 hours the first day and 1.5 hours each of the other two days, so 3 + 1.5 + 1.5 = 4.5 hours per week.  Speed = 20 / 4.5 = 4.44 mph.  #### 4.44` |

| Question: | Stephen's groceries cost $40.00.  A 25% platform fee is added, plus $3.00 delivery and a $4.00 tip. What is the final price? |
|---|---|
| Answer (GT): | 57 |
| Transition: | FALSE→TRUE |
| Original CoT: | `25% of $40.00 is $10.00.  Add delivery to the original bill: $40.00 + $3.00 = $43.00.  Then add the tip of $4.00:  $43.00 + $4.00 = $47.00.` |
| LAMP: | `25% of $40.00 is $10.00.  Add $3.00 delivery and $4.00 tip. Final = $40.00 + $10.00 + $3.00 + $4.00 = $57.00.` |

Table 2. **Mixed qualitative outcomes under self-reward (LAMP).** We show one TRUE→FALSE regression (Case 38) where local edits break global accounting, and one FALSE→TRUE correction (fees and tip) where aggregation is fixed.

changed ($\approx$ 4–6%), reinforcing that the rate of recovery is insensitive to problem difficulty, but the preservation rate collapses, leading to much weaker net accuracy.

### 3.5 QUALITATIVE ANALYSIS

We probe how self-reward reshapes reasoning by contrasting a TRUE→FALSE (TF) regression and a FALSE→TRUE (FT) correction (Table 9). In **Case 38** (*weekly pace*), the baseline correctly aggregates weekly time ($3 + 1.5 + 1.5 = 6$ h) to obtain $60/6 = 10$ mph. Under self-reward, LAMP over-edits toward a per-day normalization and mis-aggregates runtime (claimed $4.5$ h), yielding an incorrect $4.44$ mph. This TF pattern reflects a local reward preference for seemingly plausible partial computations (e.g., daily averaging) that break global constraints (total distance/time consistency).

In contrast, **Case 58** (*fees and tip*) shows a typical FT fix: the baseline omits the platform fee and reports $47; LAMP correctly aggregates base price, fee, delivery, and tip to reach the ground-truth $57.

Beyond these two cases, our broader inspection (Fig. 3) finds that self-reward frequently repairs arithmetic omissions and bookkeeping slips (FT), but can also induce TF regressions when local cues outweigh global consistency. To curb TF without suppressing FT, we rely on: *confidence gating* (edit only low-confidence tokens), *span-based selection* with locality windows, *partial-freeze (clamp) decoding* for high-confidence positions, *step-size clipping and early stop* when reward deltas are small, and a modest *edit budget*. These constraints keep edits focused where uncertainty and reward sensitivity align while preserving global accounting and units.

## 4 RELATED WORK

**Diffusion Language Models.**     Diffusion-based large language models (dLLMs) have recently emerged as strong alternatives to autoregressive models (ARMs) for text generation. Masked diffusion models such as LLaDA (Nie et al., 2025), Dream (Ye et al., 2025a), and Mercury (Labs & collaborators, 2025) generate tokens in parallel through iterative denoising and re-masking, offering advantages in decoding flexibility and bidirectional context modeling. Recent scaling efforts

(e.g., d1 (Zhao et al., 2025)) demonstrate competitive accuracy with ARMs. Nonetheless, dLLMs lag on reasoning-intensive tasks and typically require more inference steps due to the lack of KV caching (Li et al., 2025a; Liu et al., 2025b). This motivates test-time approaches that enhance reasoning without retraining.

**Inference-Time Scaling in Diffusion Models.** A growing line of work studies how to allocate extra computation at inference to improve dLLM outputs. *Search-based methods* include particle Gibbs sampling for discrete diffusion (Dang et al., 2025) and classical search strategies that combine local and global exploration (Zhang et al., 2025). *Scheduler modifications* such as ReMDM (Wang et al., 2025) introduce remasking to allow iterative error correction, while Prophet (Li et al., 2025a) leverages early convergence to commit confident tokens. Other extensions such as MDM-Prime (Chao et al., 2025) insert intermediate token states to reduce idle steps. These methods primarily target fluency or efficiency, leaving a gap in reasoning-specific adaptation.

**Test-Time Reasoning in Language Models.** For autoregressive LMs, several approaches exploit additional inference compute to improve reasoning. Chain-of-thought prompting (Wei et al., 2022), self-consistency (Wang et al., 2022), and verifier-guided search (Yao et al., 2023a) enhance reasoning by reranking or aggregating multiple trajectories. Most relevant is LatentSeek (Li et al., 2025b), which showed that treating hidden states as optimizable latents and updating them with policy gradients can significantly improve reasoning. However, direct transfer to diffusion fails: dLLMs lack a left-to-right causal structure and instead operate on globally masked updates. To date, no general framework exists for per-instance latent adaptation in diffusion LMs.

**Guidance and Reinforcement for Diffusion Models.** Gradient-based control has been widely explored in continuous diffusion, e.g., classifier guidance and score distillation (Ho et al., 2020; Dhariwal & Nichol, 2021). For discrete diffusion, recent work examined simple guidance strategies (Schiff et al., 2025) and reward-weighted sampling (Dang et al., 2025), but these operate on distributions or trajectories rather than per-instance latent optimization. Our work builds on these insights but introduces a diffusion-specific, instance-level framework: reward-guided *policy-gradient* adaptation on masked latents, coupled with remasking and clamp-and-inpaint decoding for global consistency.

## 5 CONCLUSION AND FUTURE WORK

We introduced **LAMP**, a training-free framework for reward-guided latent adaptation in masked diffusion language models. By treating hidden token states as editable latents, applying sparse policy-gradient updates, and constraining re-decoding through clamp-and-inpaint, LAMP improves reasoning accuracy at test time without modifying model parameters. Experiments across GSM8K, MATH-500, and AIME2024 show consistent gains on multiple dLLM backbones, highlighting the value of aligning diffusion's revisable decoding process with targeted reward feedback.

**Future directions.** Several promising avenues remain open for exploration. First, richer forms of supervision could be incorporated. Current experiments rely primarily on outcome-based self-reward, which provides only a sparse binary signal. Extending to *process supervision* that evaluates intermediate reasoning steps—or leveraging verifiers trained to detect local consistency—could enable the adaptation process to align more closely with logical correctness and to correct errors earlier in the reasoning trajectory. Second, LAMP could be extended beyond single-turn adaptation to *interactive or multi-turn settings*, where reward feedback is provided iteratively, potentially augmented by retrieval systems, symbolic solvers, or external critics. Such settings may be particularly valuable for long-horizon reasoning tasks or program synthesis, where one-shot reward is often insufficient. Finally, future work could explore adaptation beyond language, applying the same latent-policy principle to multimodal diffusion models where structured feedback is available, such as grounded reasoning in vision-language settings or structured prediction tasks in science and engineering domains.

Overall, LAMP demonstrates that reward-guided latent optimization provides a simple yet effective axis for advancing the reasoning capabilities of diffusion language models, complementing both autoregressive prompting strategies and emerging inference-time scaling methods.

**Ethics Statement.** This work investigates test-time *reasoning* adaptation for masked diffusion LMs on public, non-sensitive math datasets (e.g., GSM8K, MATH-500, AIME 2024) and does not involve human subjects, private information, or proprietary data; IRB approval was not required. Our Perfect Sparse Reward Model (PSRM) uses only instance-local ground-truth answers to compute binary correctness and does not alter model weights. We will not redistribute third-party checkpoints and will respect their original licenses; released code/configs will include usage guidelines discouraging deployments that could violate academic integrity or safety policies. While the technique could in principle be repurposed to optimize undesirable behaviors, our experiments are task-constrained, and we recommend domain-appropriate safety filters for broader applications. Environmental impact is limited: LAMP is training-free and adds modest inference overhead. The authors declare no conflicts of interest; sources of support will be disclosed per venue policy.

**Reproducibility Statement.** We have made every effort to ensure the reproducibility of our results. All datasets used in our experiments are publicly available and are described in detail in Section 3. Preprocessing steps and evaluation metrics are documented in Appendix B. Our implementation, including training and evaluation scripts, is provided as anonymized supplementary material. Hyperparameters and experimental settings are reported in Appendix Section D. Together, these resources allow independent researchers to replicate our findings and extend our work.

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

# SUMMARY OF THE APPENDIX

This appendix provides additional details for the ICLR 2026 submission titled ***LAMP: Latent Adaptation via Masked Policy for Diffusion Language Models***. It is organized as follows:

- §A: **LLM Usage**.
- §B: **Datasets, Preprocessing, and Prompt Formats**.
- §C: **Implementation Details** and PyTorch-style **Pseudo-code**.
- §D: **Hyperparameters** and per-run **Configurations**.
- §E: **Qualitative Examples**.

## A   LLM USAGE

In preparing this work, we used large language models only as auxiliary tools for grammar refinement, code formatting, and literature search. No LLM was used to generate research ideas, design experiments, or analyze results. All conceptual contributions were developed independently by the authors.

## B   DATASETS, PREPROCESSING, AND PROMPT FORMATS

**Benchmarks.**   We adopt the LatentSeek-style evaluation protocol on three mathematical reasoning datasets. **GSM8K** (Cobbe et al., 2021) contains 8,500 grade-school math word problems; we evaluate on the official test split of 1,319 questions. **MATH-500** (Hendrycks et al., 2021) is a curated 500-problem subset covering algebra, geometry, number theory, and calculus. **AIME 2024** (Zhang et al., 2024) comprises 30 questions from the 2024 American Invitational Mathematics Examination. All evaluations are zero-shot on the official splits.

**Prompt styles.**   We use two prompting styles. *Type 1* requests only the final boxed answer. *Type 2* requests step-by-step reasoning (rationale) followed by the boxed answer. In both cases we enforce `\boxed{}` to ease parsing.

**Prompt templates (compact blocks).**   To avoid wide tables and incompatible verbatim-in-table issues, we present prompts as narrow, monospaced blocks that line-wrap gracefully.

*GSM8K (Type 2).*

```
System:  You are a precise math question solver.  Solve this
math problem.
User:  QUESTION: {q} Let's think step by step.  Please
provide your thought process and your final answer
separately and respond in JSON with keys thought process and
final answer.  For example:  { "thought process":  "...",
"final answer":  "..." }.  Note:  the final answer must be a
pure number without units or explanation.
```

*MATH-500 / AIME 2024 (Type 2).*

```
System:  You are a precise math question solver.  Solve this
math problem.
User:  QUESTION: {q} Let's think step by step.  Please
provide your thought process and your final answer
separately and respond in JSON with keys thought process
and final answer.
```

*Type 1 variant (final answer only).*

```
System:  You are a precise math answerer.
User:  QUESTION: {q} Return only the final numeric result in
```

```
\boxed{} format, e.g., \boxed{42}. Do not include steps or
extra text.
```

**Answer extraction and normalization.** We first extract the \boxed{} span; if absent, we fall back to the last numeric/string-like token sequence. Normalization includes case-folding, Unicode NFC, whitespace and thousands-separator removal, fraction simplification, rounding of decimals (six significant figures), and evaluation of simple arithmetic expressions. Exact match (pass@1) is:

$$\text{EM}(\hat{a}, a^\star) = \mathbf{1}\big[\,\text{normalize}(\hat{a}) = \text{normalize}(a^\star)\,\big].$$

**Self-reward verifiers.** For self-rewarded settings, we use lightweight rule-based checks for format and numeric validity, plus dataset-specific sanity checks. The verification prompts are short, single-purpose instructions:

*Correctness check.*

```
INSTRUCTIONS: Decide if the provided answer is correct.
Output exactly one token: <ANS>True or <ANS>False.
```

*Calculation check.*

```
INSTRUCTIONS: (1) Extract all calculations; (2) recompute
them independently; (3) compare with the solution. If any
discrepancy, output False; else True.
```

*Understanding check.*

```
INSTRUCTIONS: Verify that the reasoning interprets the
problem correctly and answers the asked quantity. Return
True if aligned; otherwise False.
```

*Completeness check.*

```
INSTRUCTIONS: Verify that a final, explicit numeric answer
is provided (not just a formula). Return True or False.
```

## C  IMPLEMENTATION DETAILS AND PSEUDO-CODE

**Environment.** All experiments use PyTorch with CUDA 12.x. Backbones: **LLaDA-8B**, **LLaDA-1.5**, and **Dream-7B**. Random seed $42$; deterministic CuDNN where available. Adaptation is per-instance; no gradient accumulation.

**Decoding.** We use each model's native masked-denoising scheduler and early-commit heuristics (if provided). Sampling temperature $= 1.0$; no top-$k$ or nucleus sampling.

**LAMP defaults.** Edit budget $k = 10\%$ (by lowest confidence), policy-gradient steps $K = 2$, learning rate $\eta = 0.3$, trust-region regularization $(\lambda_{\text{KL}}, \lambda_2) = (0.1, 0.05)$, confidence-gating $(\tau, \varepsilon) = (0.6, 0.05)$.

PYTORCH-STYLE PSEUDO-CODE (MINIMAL DEPENDENCIES)

We avoid external code environments; the snippet compiles as plain text and can be implemented directly.

```
# LAMP: Latent Adaptation via Masked Policy
def LAMP_decode(model, prompt, reward_fn, k=0.1, K=2,
eta=0.3,
  tau=0.6, eps=0.05, lam_kl=0.1, lam_l2=0.05):
 # 1) Baseline decode with hidden states and logits
 y0, h0, q0 = model.diffuse(prompt, return_hidden=True)
 conf = q0.max(dim=-1).values
```

| methods | model | max len | prmpt idx | #GPU | lr | opt | $\rho$ | dtype | steps |
|---|---|---|---|---|---|---|---|---|---|
| LAMP (SELF) | LLaDA-8B | 1024 | 1 | 1 A100 | 0.3 | Adam | 0.1 | bf16 | 10 |
| LAMP (SELF) | LLaDA-8B | 1024 | 2 | 1 A100 | 0.3 | Adam | 0.1 | bf16 | 10 |
| LAMP (SELF) | LLaDA-1.5 | 1024 | 1 | 1 A100 | 0.3 | Adam | 0.1 | bf16 | 10 |
| LAMP (SELF) | LLaDA-1.5 | 1024 | 2 | 1 A100 | 0.3 | Adam | 0.1 | bf16 | 10 |

Table 3. Run configurations for **LAMP (Self)** on GSM8K.

| methods | model | max len | prmpt idx | #GPU | lr | opt | $\rho$ | dtype | steps |
|---|---|---|---|---|---|---|---|---|---|
| LAMP (PSRM) | LLaDA-8B | 1024 | 1 | 1 A100 | 0.3 | Adam | 0.1 | bf16 | 10 |
| LAMP (PSRM) | LLaDA-8B | 1024 | 2 | 1 A100 | 0.3 | Adam | 0.1 | bf16 | 10 |
| LAMP (PSRM) | LLaDA-1.5 | 1024 | 1 | 1 A100 | 0.3 | Adam | 0.1 | bf16 | 10 |
| LAMP (PSRM) | LLaDA-1.5 | 1024 | 2 | 1 A100 | 0.3 | Adam | 0.1 | bf16 | 10 |

Table 4. Run configurations for **LAMP (PSRM)** on GSM8K.

```
S = conf.argsort()[:  int(k * len(conf))] #
lowest-confidence
z = h0[S].detach().clone().requires_grad_(True)
baseline = 0.0

for t in range(K):
  q = torch.softmax(model.head(z), dim=-1)
  y_tilde = q.multinomial(1).squeeze(-1)
  y, h, q_new = model.diffuse(prompt, fixed={S: y_tilde},
return_hidden=True)
  r = reward_fn(y); baseline = 0.9*baseline + 0.1*r
  logprob = torch.log(q[torch.arange(len(S)),
y_tilde]).sum()
  pg_loss = - (r - baseline) * logprob
  kl_reg = torch.nn.functional.kl_div(q.log(), q0[S],
reduction="batchmean")
  l2_reg = ((z - h0[S])**2).mean()
  loss = pg_loss + lam_kl * kl_reg + lam_l2 * l2_reg
  g, = torch.autograd.grad(loss, z); z = z - eta * g

final_conf = torch.softmax(model.head(z),
dim=-1).max(dim=-1).values
mask = (final_conf >= tau) & ((final_conf - conf[S]) >=
eps)
fixed = { int(S[j]):  int(y_tilde[j]) for j in
torch.where(mask)[0] }
y_star, _, _ = model.diffuse(prompt, fixed=fixed)
return y_star
```

## D  HYPERPARAMETERS AND RUN CONFIGURATIONS

**Global defaults.**  We fix hyperparameters across experiments; beyond light sanity checks on 20-dev subsets, no broad sweeps. Adam optimizer; trust-region coefficient $\rho{=}0.1$; bf16 precision; **10** diffusion refinement steps; maximum output length **128** tokens. Edits target the *answer span*; rationales are refined indirectly via masked denoising.

## E  QUALITATIVE EXAMPLES

**Analysis.**  Arithmetic aggregation cases (groceries, stories, annuities) benefit from revising low-confidence tokens and re-sampling consistent totals. Regressions arise when confident but incorrect local edits disrupt global consistency (running speed, puzzle) or when partial functional recurrences

| methods | model | max len | prmpt idx | #GPU | lr | opt | $\rho$ | dtype | steps |
|---------|-------|---------|-----------|------|-----|-----|--------|-------|-------|
| LAMP (SELF) | LLaDA-8B | 1024 | 1 | 1 A100 | 0.3 | Adam | 0.1 | bf16 | 10 |
| LAMP (SELF) | LLaDA-1.5 | 1024 | 2 | 1 A100 | 0.3 | Adam | 0.1 | bf16 | 10 |
| LAMP (SELF) | Dream-7B | 1024 | 1 | 1 L40S | 0.3 | Adam | 0.1 | bf16 | 10 |
| LAMP (SELF) | Dream-7B | 1024 | 2 | 1 L40S | 0.3 | Adam | 0.1 | bf16 | 10 |

Table 5. Run configurations for **LAMP (Self)** on MATH-500.

| methods | model | max len | prmpt idx | #GPU | lr | opt | $\rho$ | dtype | steps |
|---------|-------|---------|-----------|------|-----|-----|--------|-------|-------|
| LAMP (PSRM) | LLaDA-8B | 1024 | 1 | 1 A100 | 0.3 | Adam | 0.1 | bf16 | 10 |
| LAMP (PSRM) | LLaDA-1.5 | 1024 | 2 | 1 A100 | 0.3 | Adam | 0.1 | bf16 | 10 |
| LAMP (PSRM) | Dream-7B | 1024 | 1 | 1 L40S | 0.3 | Adam | 0.1 | bf16 | 10 |
| LAMP (PSRM) | Dream-7B | 1024 | 2 | 1 L40S | 0.3 | Adam | 0.1 | bf16 | 10 |

Table 6. Run configurations for **LAMP (PSRM)** on MATH-500.

are overextended (functional equation). Self-rewarded latent updates improve robustness but require careful regularization and gating to avoid over-corrections.

| methods | model | max len | prmpt idx | #GPU | lr | opt | $\rho$ | dtype | steps |
|---|---|---|---|---|---|---|---|---|---|
| LAMP (SELF) | LLaDA-8B | 1024 | 1 | 1 A100 | 0.3 | Adam | 0.1 | bf16 | 10 |
| LAMP (SELF) | Dream-7B | 1024 | 2 | 1 L40S | 0.3 | Adam | 0.1 | bf16 | 10 |

Table 7. Run configurations for **LAMP (Self)** on AIME 2024.

| methods | model | max len | prmpt idx | #GPU | lr | opt | $\rho$ | dtype | steps |
|---|---|---|---|---|---|---|---|---|---|
| LAMP (PSRM) | LLaDA-8B | 1024 | 1 | 1 A100 | 0.3 | Adam | 0.1 | bf16 | 10 |
| LAMP (PSRM) | Dream-7B | 1024 | 2 | 1 L40S | 0.3 | Adam | 0.1 | bf16 | 10 |

Table 8. Run configurations for **LAMP (PSRM)** on AIME 2024.

| | |
|---|---|
| **Question** | John runs 60 miles a week on 3 days. He runs 3 hours on day 1 and half as much on the other two days. How fast does he run? |
| **GT** | 10 |
| **Transition** | TRUE→FALSE |
| **Original CoT** | `Total time:  3 + 1.5 + 1.5 = 6.  Speed = 60/6 = #### 10.` |
| **LAMP** | `Day avg 20 miles; time 4.5 hours; 20/4.5 = #### 4.44.` |
| **Question** | Stephen's groceries cost $40. A 25% platform fee is added, plus $3 delivery and $4 tip. Final price? |
| **GT** | 57 |
| **Transition** | FALSE→TRUE |
| **Original CoT** | `Mis-adds:  40+3+4=47.  #### 47.` |
| **LAMP** | `25% of 40 is 10; total = 40+10+3+4 = #### 57.` |
| **Question** | A 1000-piece puzzle: Poppy places a quarter; mom places a third of remaining. How many left? |
| **GT** | 500 |
| **Transition** | TRUE→FALSE |
| **Original CoT** | `Poppy=250; remaining 750; mom=250; leftover #### 500.` |
| **LAMP** | `Finds 250 and 250 but outputs #### 250.` |
| **Question** | Week 1: 20, 40, 60 stories. Week 2 each doubles. Total stories? |
| **GT** | 360 |
| **Transition** | FALSE→TRUE |
| **Original CoT** | `Sums to 300.` |
| **LAMP** | `Week1=120; Week2=240; Combined=#### 360.` |
| **Question** | Deposit $20k annually for 3 years; wants $66,200 after third deposit. Minimal compound rate? |
| **GT** | 10 |
| **Transition** | FALSE→TRUE |
| **Original CoT** | `Treats as single deposit; #### 0.` |
| **LAMP** | $FV = P\frac{(1+r)^n-1}{r}$; `solve` $66200 = 20000 \cdot \frac{(1+r)^3-1}{r}$; `#### 10.` |
| **Question** | $f(x) + f(y) = f(x+y) - xy - 1$, $f(1) = 1$. Integers $n$ with $f(n) = n$? |
| **GT** | $1, -2$ |
| **Transition** | TRUE→FALSE |
| **Original CoT** | `Finds` $n=1$; `misses` $-2$. |
| **LAMP** | `Drifts; outputs extraneous #### 8.` |

Table 9. **Mixed qualitative outcomes under self-reward (LAMP).** We show regressions (TRUE→FALSE) and successful corrections (FALSE→TRUE).