# OpenReview forum: "Latent Adaptation with Masked Policy for Diffusion Language Models"
_ICLR.cc/2026/Conference — ICLR 2026 Conference Withdrawn Submission_

### Official Review · Reviewer_Qehn · 2025-10-27

**Soundness:** 2
**Presentation:** 2
**Contribution:** 2
**Rating:** 4
**Confidence:** 4

**Summary:**

This paper presents Latent Adaptation via Masked Policy (LAMP), a training-free framework designed to enhance the reasoning capabilities of discrete diffusion language models (dLLMs) during test time. LAMP identifies low-confidence tokens from a baseline decoding and applies reward-guided policy-gradient updates to their latent representations to maximise the expected reward. Empirically, LAMP achieves strong performance across various dLLMs and reasoning tasks.

**Strengths:**

- The proposed LAMP method is straightforward and intuitive. The idea of updating the final latent representation by maximising the expected reward is simple yet effective.
- LAMP demonstrates substantial and consistent improvements across multiple dLLM backbones (LLaDA, DREAM) and challenging mathematical reasoning benchmarks (GSM8K, MATH-500, AIME 2024).

**Weaknesses:**

- My main concern lies in the clarity and coherence of the paper’s narrative. The presentation of the algorithm details, such as the CONSTRAINEDDIFFUSE operation and baseline values, is either missing or insufficiently explained, making it challenging to understand how the sampling procedure operates fully.
- The overall intuition makes sense to me. Introducing an additional gradient update on the output representation to maximise the expected reward should indeed improve performance. However, this raises two concerns:
    - Choice of reward function: The paper uses the PSRM reward, which depends on the ground-truth answer. Since the ground truth is not available at test time, this choice makes the approach somewhat confusing. It would also be valuable to compare the effects of different reward models.
    - Increased computational cost: The additional refinement steps inevitably increase computation. The paper should discuss or quantify this overhead and analyse the trade-off between performance gains and efficiency.

**Questions:**

- The overall algorithm is unclear:
    - What are the operations DIFFUSE and CONSTRAINEDDIFFUSE
    - What does the baseline $b$ represent? Do you use releave-one-out or other variance reduction tricks?
    - What is $R_{stab}$
    - what is $\mathcal{L}_{PG}$. It seems that you are applying policy gradient to maximise the expected reward, but in line 8 of Algorithm 1, there is an additional $R_{\text{stab}}$. Could you clarify its purpose?
- The definition of the PSRM reward depends on the ground-truth answer. However, since the ground truth is not available in practice, how do you compute this reward?
- I recommend including an additional baseline: instead of updating only the latents of low-confidence positions, update the latents of all positions using the proposed policy gradient. This comparison could help clarify whether the selective update strategy truly contributes to performance improvements, or if the observed gains primarily come from the general policy gradient refinement itself.

---

### Official Review · Reviewer_Rocd · 2025-10-31

**Soundness:** 2
**Presentation:** 3
**Contribution:** 1
**Rating:** 2
**Confidence:** 2

**Summary:**

This paper proposes LAMP (Latent Adaptation via Masked Policy), a training-free framework for test-time reasoning improvement in diffusion language models (dLLMs). Unlike autoregressive models, dLLMs decode in parallel through iterative denoising, enabling latent-space intervention. LAMP performs reward-guided policy-gradient updates on a small subset of low-confidence token latents, followed by a clamp-and-inpaint step that preserves accepted edits while re-inpainting remaining tokens for global coherence. The method supports both lightweight self-reward signals and a Perfect Sparse Reward Model (PSRM) that provides binary correctness feedback.

**Strengths:**

The methodology is well-clarified. The authors present clear algorithmic descriptions, including pseudo-code and ablation studies that isolate the impact of sparse selection, reward design, and confidence gating. Experiments are comprehensive, covering diverse backbones (LLaDA, LLaDA-1.5, Dream).

**Weaknesses:**

- Insufficient Citation of Foundational Diffusion Model Works: As a paper discussing diffusion language models (specifically, masked diffusion models), the literature review and citation section is incomplete. At a minimum, the foundational papers that introduced the mask (absorbing) discrete diffusion model [1] and subsequent mainstream model structures and loss functions[2,3,4] should be cited and discussed to properly contextualize the methodology.

- Over-reliance on PSRM for Substantial Gains: The most significant limitation is that all claimed substantial performance improvements rely entirely on the PSRM. PSRM requires access to the ground-truth answer during test-time inference to provide a binary reward signal. This is infeasible in any real-world deployment scenario.

[1] Structured Denoising Diffusion Models in Discrete State-Spaces, arXiv:2107.03006

[2] Simplified and Generalized Masked Diffusion for Discrete Data, arXiv.2406.04329

[3] Your Absorbing Discrete Diffusion Secretly Models the Conditional Distributions of Clean Data, 2406.03736

[4] Simple and Effective Masked Diffusion Language Models, arXiv.2406.07524

**Questions:**

Practical Reward Modeling: Given the impracticality of PSRM, what are the authors' thoughts on using a trained discriminative verifier (Reward Model) as a reward source, and how would they anticipate its performance would compare to the Self-Reward heuristic?

---

### Official Review · Reviewer_VtTH · 2025-11-01

**Soundness:** 2
**Presentation:** 1
**Contribution:** 2
**Rating:** 2
**Confidence:** 4

**Summary:**

This paper presents LAMP (Latent Adaptation via Masked Policy), a novel training-free framework designed to enhance the reasoning capabilities of masked diffusion language models (dLLMs) at test time. Its core contribution lies in a per-instance adaptation method that first identifies a sparse set of low-confidence tokens from an initial decoding step. It then applies a few steps of policy gradient updates directly to the hidden latents corresponding to those tokens, guided by a reward signal—which can range from a lightweight self-reward mechanism to a Perfect Sparse Reward Model (PSRM) for oracle-level supervision. The process concludes with a “clamp-and-inpaint” decoding strategy: high-confidence edits are fixed, while the diffusion model’s inherent bidirectional modeling capability is leveraged to reinpaint the rest of the sequence, thereby maintaining global coherence.  Empirically,  LAMP shows consistent improvements in accuracy on several challenging mathematical reasoning benchmarks, including GSM8K, MATH-500, and AIME across multiple dLLM backbones.

**Strengths:**

- The idea of test-time adaptation combined with diffusion language models is novel and interesting.

**Weaknesses:**

1. The writing quality requires significant improvement. Upon my initial reading, I found several parts of the paper confusing. The use of certain informal expressions—such as "sizable jumps" (Line 292)—undermines the academic tone. Additionally, the overreliance on bullet points in nearly every section (e.g., Section 2.1, "Diffusion Decoding" and "Inference Characteristics") disrupts the narrative flow and reduces readability. A more coherent and professionally articulated presentation is needed.
2. As I am not deeply familiar with test-time scaling, one conceptual concern arises with the use of the Perfect Sparse Reward Model (PSRM): since it provides oracle-level supervision during inference, does the resulting improvement in accuracy remain meaningful? Intuitively, performance gains are almost guaranteed under such a setting, which may limit the practical significance of the findings.
3. From the current exposition, it is unclear what the real-world inference-time utility of LAMP is. The authors should better clarify the practical implications and applicability of their method.
Overall, the paper in its current version leaves considerable room for improvement, particularly in writing clarity and articulating the broader impact of the proposed framework.

**Questions:**

1. Related to Weakness 1: What is the justification for using PSRM as a form of supervision, especially given that ground-truth answers are unavailable in real inference scenarios? How does this choice affect the validity and generalizability of the reported results?
2. Regarding Figure 2: When the iteration number is 1, does this indicate one inference step followed by one edit? If so, the results appear inconsistent with prior work—for instance, LLaDA typically performs poorly on GSM8K with fewer than 16 inference steps. How does LAMP achieve strong performance (over 70%) with only two inference steps?
3. In Table 1, the results on GSM8K and MATH-500 seem notably lower than those reported in the original LLaDA and LLaDA 1.5 papers. For example, LLaDA was reported to achieve 78% on GSM8K, while Table 1 shows only 71.3%. What explains this discrepancy?
4. Have the authors considered designing more effective self-reward mechanisms? Self-reward appears more meaningful than PSRM in practice, as it does not rely on ground-truth answers, making it better suited for real-world inference.

---

### Official Review · Reviewer_jqoK · 2025-11-02

**Soundness:** 2
**Presentation:** 1
**Contribution:** 2
**Rating:** 4
**Confidence:** 4

**Summary:**

This paper introduces **LAMP (Latent Adaptation with Masked Policy)**, a training-free framework for instance-level test-time adaptation in masked diffusion language models (DLMs). LAMP treats hidden token states as editable latents and applies one or two policy-gradient updates guided by reward signals, followed by a clamp-and-inpaint decoding step that propagates edits through the diffusion process. The rewards can be lightweight self-rewards (e.g., format or consistency checks) or stronger outcome-based signals such as the Perfect Sparse Reward Model (PSRM) that knows the answer. Experiments show that all components: sparse selection, reward choice, and clamp-and-inpaint are essential for the method’s success.

**Strengths:**

The experimental results are comprehensive and provide valuable insights into the effect of latent adaptation through masked policy optimization. The proposed framework demonstrates clear performance gains, with the **Perfect Sparse Reward Model (PSRM)** yielding substantial improvements over the base models. However, these results should be interpreted with caution, as PSRM leverages access to ground-truth answers, which may introduce gradient leakage and lead to an unfair advantage in evaluation.

**Weaknesses:**

1. **Figure Reference:** Figure 1 is never referenced in the introduction. The authors should ensure all figures are properly introduced and discussed in the main text.

2. **Line 41:** Missing citation in “Recent systems such as LLaDA, Dream, Mercury, and d1 scale competitively…” Please provide references for these systems.

3. **Equation 3:** The variable $\hat{y}$ is not defined. Clarify its meaning to improve readability.

4. **Lines 122–125:** The description of *constrained infilling* is unclear. Please elaborate on how it is implemented and its role within the proposed framework.

5. **Equation 4:** The variable $\tilde{y}_i$ is not defined. A clear definition is needed for consistency in notation.

6. **Line 201:** The source or rationale for the chosen values of $\tau$ and $\epsilon$ is not provided. Clarify how these hyperparameters were determined.

7. **Acronym Consistency:** The acronym **PSRM (Perfect Sparse Reward Model)** is redefined multiple times throughout the paper. It should be defined once at its first occurrence and used consistently thereafter. Otherwise, it seems like the authors want to introduce a new acronym.

8. **Evaluation of Self-Reward:** In most practical scenarios, the reward model will not have access to ground truth, making self-reward the natural choice for the proposed method LAMP. However, the results show only marginal improvements over the baselines in this setting, which weakens the overall benefit of latent adaptation. The authors are encouraged to provide further analysis or justification for this limited improvement and the intended use cases.

**Questions:**

Please see the weaknesses above.

---

### Note · Authors · 2025-11-12

I have read and agree with the venue's withdrawal policy on behalf of myself and my co-authors.